# Fasting, food and farming: Value chains and food taboos in Ethiopia

**Eline D'Haene**[1]*, **Senne Vandevelde**[2], **Bart Minten**[3]

**1** Department of Plants and Crops, Ghent University, Ghent, Belgium, **2** Directorate-General for International Partnerships, European Commission, Brussels, Belgium, **3** International Food Policy Research Institute, Yangon, Myanmar

☺ These authors contributed equally to this work.
* Eline.DHaene@UGent.be

**Data Availability Statement:** Data: Habte, Yetimwork, 2019, "Dairy Value Chain Survey in Ethiopia: Producer Survey", https://doi.org/10. 7910/DVN/7AIKZH, Harvard Dataverse, V1, UNF:6: t2Vb/WuOUY3a3+tb8BA4QA== [fileUNF].

## Abstract

The impact of food taboos–often because of religion–is understudied. In Ethiopia, religious fasting by Orthodox Christians is assumed to be an important impediment for the sustainable development of a competitive dairy sector and desired higher milk consumption, especially by children. However, evidence is limited. Relying on unique data, we shed light on three major issues. First, we observe that the average annual number of fasting days that Orthodox adults are effectively adhering to is 140, less than commonly cited averages. Using this as an estimate for extrapolation, fasting is estimated to reduce annual dairy consumption by approximately 12 percent nationally. Second, farms adapt to declining milk demand during fasting by increased processing of milk into storable products–fasting contributes to larger price swings for these products. We further note continued sales of milk by non-remote farmers and reduced production–by adjusting lactation times for dairy animals– for remote farmers. Third, fasting is mostly associated with increased milk consumption by the children of dairy farmers, seemingly because of excess milk availability during fasting periods. Our results suggest that fasting habits are not a major explanation for the observed poor performance of Ethiopia's dairy sector nor low milk consumption by children. To reduce the impact of fasting on the dairy sector in Ethiopia further, investment is called for in improved milk processing, storage, and infrastructure facilities.

## Introduction

Religious traditions are often reflected in food choices as various religious groups lay out food recommendations to their adherents. Even if foods are available and accessible, individuals might not consume specific food items because of these traditions [1]. Despite the growing share of religious people worldwide [2] and the acknowledged importance of religion in shaping the structure of institutions and markets [3], the role of religion in shaping food choices, nutrition, and food value chains is still not well understood [4].

Religious practices are especially prevalent for animal-source foods (ASF) [5]. The practice of fasting, in particular, might reduce ASF consumption in two ways. First, aggregate

**Funding:** The research presented here was conducted as part of the CGIAR Research Program on Policies, Institutions, and Markets (PIM), which is led by IFPRI. This study was made possible by the generous support of the American people through USAID under the Feed the Future Innovation Lab for Livestock Systems (LSIL), which is implemented by the Institute of Food and Agricultural Sciences of the University of Florida in partnership with the International Livestock Research Institute (ILRI). LSIL is funded by the United States Agency for International Development (USAID) through a five-year Leader with Associates Cooperative Agreement Award No. AID-OAA-L-15-00003. The funder had no role in study design, data collection, and analysis, decision to publish, or preparation of the manuscript.

**Competing interests:** The authors have declared that no competing interests exist.

consumption levels might be directly affected as adherents are not allowed to consume ASF during fasting periods. Second, fasting might result in lower availability and higher pricing of ASF over the long term due to lower investments and broader value chain effects. This reduction is unfortunate, as there is a strong link between the consumption of (a reasonable amount of) ASF and improved nutritional outcomes in low income countries [6]. By contrast, long-term consumption of increasing amounts of red and processed meats has been linked to a higher risk for heart disease, cancer, diabetes and premature death in upper middle and high income countries [7–9].

We study the effects of fasting practices, focusing on the case of dairy production and consumption in Ethiopia, the second most populous country in Africa. Ethiopian diets are mainly composed of cereals (especially teff), roots, tubers and pulses. The consumption of fruit and vegetables as well as ASF are low and cereals remain the major contributors to dietary energy and protein [10]. Annual consumption of ASF per adult equivalent vary around 20 kg, of which dairy products represent 71%, beef 16% and mutton and goat 6%. Chicken, honey, eggs and fish products are less frequently consumed whereas pork is rarely consumed because of religious customs [11].

We assess the effects of seasonal fasting rituals embedded within the Ethiopian Orthodox community, the dominant religious group in the country. During several fasting periods annually, Orthodox members are to refrain from consuming ASF and therefore adhere to vegan dietary patterns throughout the entire day. Additionally, observants are required to refrain from eating or drinking before noon or the Liturgy and take no more than one meal a day during these fasting periods [12, 13]. Fasting (and associated feasts) occur at different times throughout the year thereby creating considerable demand swings for ASF. This makes Ethiopian diets not only susceptible to agroclimatic patterns, but also to the sequential cycle of religious fasting. Studying this issue in Ethiopia is an important topic given low overall consumption of milk [14], especially by children, and the potential prohibitive effects this has on investments in dairy production, possibly contributing to the underdevelopment of the country's dairy sector compared to other countries in the region [15].

We focus in this analysis on three research questions in particular. First, we evaluate the number of effective fasting days adhered to by Orthodox members. Fasting periods differ in length and cover both one-day fasts and longer fasting periods (16 to 55 days). However, as not all Orthodox Christians adhere to all fasts, it is difficult to assess the exact influence of fasting on Ethiopia's dairy sector. Several estimates on the number of fasting days annually can be found in the literature, with ranges of from 166 to 180 days [11, 16], and from 180 to 250 days [14, 17] being cited. With unique household member-specific fasting data at our disposal, we are able to evaluate effective fasting practices, as well as beliefs. This is important to better understand demand swings and their importance for dairy markets.

Second, we look at what the effects of these demand changes due to fasting are on the dairy value chain and, more specifically, which adaptation strategies milk producing households develop to overcome the effects of low demand during fasting periods. Existing–mostly qualitative–studies show important effects of fasting on the value chain. A reduced, or even a complete lack of, market access as well as lower dairy prices during fasting have been found at the farm level [18, 19]. There is, however, only limited quantitative evidence on the value chain effects of fasting and on adjustments made by producing households. Moreover, no attention has been paid to differential effects by the degree of market access of milk producing households.

Third, we assess the impact of fasting practices on children's milk consumption. Previous studies have illustrated negative repercussions of fasting on dietary diversity and intake of ASF for children, who are, in principle, exempt from fasting [20, 21]. In contrast to previous studies, we test the impact of fasting on milk intake of children within milk producing households,

who are less affected by availability issues. We do so by relying on detailed consumption data collected in a survey that was partly rolled out during and outside of a major fasting period.

We find that the Orthodox adult fasts on average 140 days per year and that annual milk consumption at the national level is reduced by about 12 percent because of Orthodox fasting practices. While effects in specific periods and locations might obviously be higher, this result suggests that fasting is only a partial explanation for the low dairy consumption in Ethiopia. Moreover, we find that price effects of fasting on milk are much smaller than reported in other studies, although we observe larger price swings for processed milk products. Several adaptation strategies are followed by milk producing households to reduce the economic impact of declining dairy demand during fasting, including increased processing of milk into storable milk products, reduced production (by adjusting lactation times of dairy animals) for remote farmers, and continued sales of milk for farmers with good market access. Finally, despite some children being affected by fasting practices, their number is found to be small. Based on detailed milk consumption data, we find no significant declines in milk consumption for young children during fasting. Fasting is actually found to be beneficial for most young children in milk producing households. Excess milk increases in these producing households during fasting periods, hence improving its availability for consumption by children in the household.

Our findings have several important implications. First, as the impact of fasting at the national level is found to be relatively small and as we find that children consume milk when it is available, even in fasting periods, this suggests that other issues, such as availability and affordability, and not fasting, are the main impediments to increased dairy consumption. This finding is corroborated by the high income and price elasticities for dairy products [11]. Further investments to stimulate the dairy sector are therefore needed to increase availability of dairy products at lower prices. In contrast to the rapid growth in crop output and productivity recorded in Ethiopia over the past 20 years, ASF output has grown slowly and productivity has stagnated. This is seemingly due to low availability and adoption of improved inputs in the dairy sector leading to high and increasing prices for dairy products [22].

Second, we find that some young children participate in fasting, even if the share is relatively small. Further efforts in improved information dissemination on the potential adverse developmental effects of fasting on children is therefore needed.

Third, to help smoothen the effect of seasonal demand swings and possibly increase returns to investments in the sector, further efforts are needed towards enhancing processing practices (such as ultra-heat treated and powdered milk), ensuring greater availability of chilling centers,

**Table 1. Overview of the obligatory fasting events and feasts in Ethiopia according to the Orthodox Church doctrine.**

| Fasts | Timing |
|---|---|
| Advent fast | 40 days, from November 28 –January 6 |
| Epiphany fast | Fast on the eve of Epiphany |
| Nineveh fast | Three days (Monday to Wednesday), two weeks before the start of Lent |
| Lent fast (Easter fast) | 55 days, starts on Monday, movable start (between February 8 –March 14) |
| Sene fast (Apostles fast) | 10–40 days, starts Monday after Pentecost and ends July 12 |
| Felseta fast (Assumption fast) | 15 days, from August 7 –August 21 |
| Weekly fasting | Every Wednesday and Friday (except for the period between Easter and Pentecost) |

Source: Compiled by authors, based on Knutsson and Selinus [12], Ware [24], and The Ethiopian Orthodox Tewahedo Church [25]

as well as improving market access and transportation facilities to assure market integration and allow marketing to areas where fasting is less prevalent.

## Background

Ethiopia hosts the largest Orthodox community outside Europe. In the last census in 2007, it was estimated that the community made up 43 percent of the total population in Ethiopia [23]. Fasting in the Orthodox community entails abstinence from ASF during several official fasting periods, spread throughout the year. During those periods, no consumption of meat, eggs, or dairy products is allowed. Fasting periods consist of both one-day fasts and longer fasting seasons that often precede holy events. An overview of the fasting periods is given in Table 1.

The sequential cycle of Orthodox fasting makes demand for ASF products to vary considerably in the country, as has been shown in a number of studies. Hirvonen, Taffesse [26] found that diets are less diversified during the Orthodox Lent and Advent fasting periods, due to reduced intake of ASF, both for rural and urban households. Likewise, Abegaz, Hassen [11] mapped seasonal per capita consumption of ASF and observed that drops and peaks in intake overlap to a large extent with Orthodox fasting and feasts.

These demand swings caused by fasting affect all actors along the dairy value chain, i.e. retailers, processors, and producers. Many processors cut down their capacity during fasting periods, with reductions of 25 percent being reported [27]. Moreover, they try to limit the supply of milk during fasting using quota systems, paying lower prices, or requiring a higher milk quality from their suppliers [28]. Some processors adapt during the fasting period by building up stocks of processed milk products that they can sell outside the fasting period, while others transport their products to areas where fasting is less prevalent [27]. Since end consumers, retailers, and processors all limit the quantity of milk they purchase, fasting necessarily also affects the dairy farmers. Many farmers indicate a reduced or even a complete lack of market access, lower sales, reduced dairy prices, and increased processing of milk into less perishable products, like butter or cheese, during fasting [19, 29].

Other studies have examined the specific impact of fasting on the milk intake of young children [20, 21, 30] and on lactating mothers [31] in Ethiopia. While pregnant and lactating women and children below the age of seven years in principle are exempt from fasting [12], all of these studies found negative repercussions of Orthodox fasting practices on dietary diversity and intake of ASF. As such, Desalegn, Lambert [21] who compared dietary intake in and outside Lent fasting for children aged 6–23 months old, found that while there was no significant difference in the consumption of 7 food groups (i.e. grains, roots and tubers; pulses and nuts; other fruits and vegetables; dairy products; flesh foods; eggs; vitamin A-rich fruits and vegetables) for children of fasting mothers during and outside the Lent fasting period, a significantly higher proportion of children from non-fasting mothers was consuming grains, roots and tubers, pulses and nuts, other fruits and vegetables and eggs outside the Lent fasting period resulting in a significantly higher proportion of children of non-fasting mothers reaching the minimum acceptable diet and minimum dietary diversity criteria outside Lent than during Lent. This study thus shows that maternal fasting adherence influences children's dietary intake and nutritional status significantly, even outside of official fasting periods. Furthermore, Desalegn, Lambert [31] observed when comparing diets of fasting and non-fasting lactating mothers, that the proportion of lactating (fasting and non-fasting) mothers who consumed ASF was significantly lower during Lent than outside Lent, while a lower proportion of non-fasting mothers was consuming dark green leafy vegetables outside Lent as compared to during Lent fasting. No significant difference was found in the proportion of fasting lactating mothers who consumed pulses, dark green leafy vegetables, vitamin A-rich fruits and

vegetables and other vegetables groups between Lent and outside Lent and the proportion of non-fasting mothers consuming pulses and other vegetables groups. Kim et al. [30], who studied ASF consumption by children aged 6 to 23 months during Lent, observed that only a quarter of surveyed children consume ASF during the Lent fasting period, even though 80 percent of the surveyed households reported owning livestock. They found that caregivers are reluctant to feed ASF to their children during Lent, even when livestock products are available, because they fear the disapproval of their neighbors or the contamination of cooking utensils with non-fasting foods. A recent study by Potts, Mulugeta [32] even found that Muslim children had a higher probability (20%) of having consumed ASF as compared to children from Orthodox households and households adhering to other religions.

These trends are worrisome given low dietary diversity and low ASF consumption in Ethiopia, even outside of fasting periods. According to the 2016 Ethiopian Demographic and Health Survey, only 14 percent of the children aged 6 to 23 months received diets that met the recommended minimum dietary diversity score. Children under the age of two years consume little meat, fish, or poultry: only 8 percent for breastfeeding children and 14 percent for non-breastfeeding children. Although milk intake is somewhat higher, it is still low at 13 percent and 24 percent, respectively [33]. The low inclusion of ASF in children's diets is believed to be an important driver of the high prevalence of malnutrition in sub-Saharan Africa [34]. Yet, it has been shown that household cow ownership increases children's milk consumption and promotes linear growth and reduces stunting in young children, especially so in rural areas with thin dairy markets [6]. These studies demonstrate how Orthodox fasting practices might affect consumption and value chains of ASF. Yet, quantitative analyses of these effects remain limited–a research gap we address here.

## Conceptual framework

To conceptualize the impacts of fasting, we rely on a simple demand and supply framework (Fig 1). We first look at the results of a one-commodity analysis and then expand this to two commodities, allowing for the possibility of processing liquid milk into butter.

In a one-commodity framework (liquid milk), prices and quantity consumed meet at an equilibrium price ($P_{NF}$) and quantity ($Q_{NF}$) during non-fasting periods (Fig 1, left side). Abstinence from dairy products by some people in Ethiopia during the fasting period leads to a shift downward from $D_{NF}$ to $D_F$ in the overall demand curve for milk. If supply stays constant, prices of milk will drop to $P_F$, and it will therefore become more accessible for those people that do not fast. For the latter group, consumption might therefore go up during the period of fasting. It is also possible that supply is adjusted down with the decline in demand during fasting periods, since lactation periods in dairy animals can be planned. In that case, the price declines associated with fasting periods might be smaller. However, as lactation periods cannot easily be adjusted for relatively short fasting periods, such effects will likely be relatively small and will only exist for longer fasts. Effects of fasting on prices and quantities consumed will depend on respective demand and supply elasticities as well as the share of the population that is affected by fasting, as that will affect the size of the shift.

If a second commodity is brought into the analysis and we allow for the fact that milk can be transformed to a storable product, e.g., butter, fasting might lead to a number of different effects in the milk and butter markets. This situation is visualized by red supply curves in Fig 1. In this case, on top of demand shifts, there is a shift in the supply curve of liquid milk–from $S_{NF}$ to $S'_F$–as well. While consumption is significantly reduced, price effects are much smaller as seen in the smaller differences between ($P_{NF}$-$P'_F$) than ($P_{NF}$-$P_F$). On the other hand, effects of fasting in the butter market will be magnified. We see a supply shift to the right because of

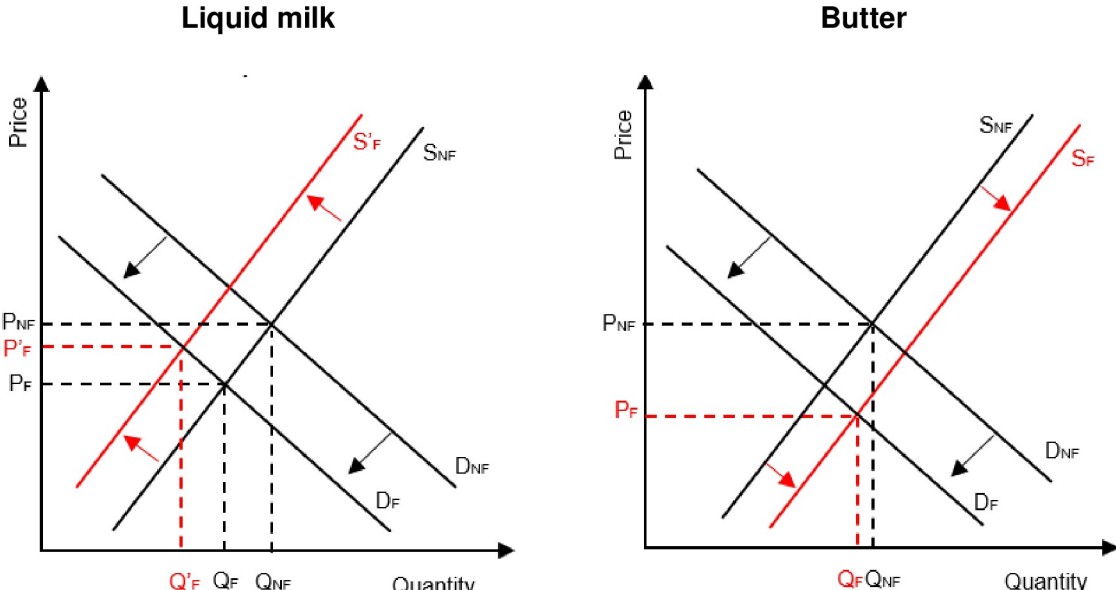

**Fig 1. Demand and supply framework illustrating the impact of fasting on the liquid milk and butter markets.** (1) Lent; (2) Easter; (3) Easter-May; (4) Jun-Advent; (5) Advent; (6) Christmas; (7) Christmas-current.

the extra processing of liquid milk into butter during fasting. As the demand for this commodity is also affected by fasting habits, we further see a shift of the demand curve downwards, putting extra downward pressure on butter prices, which are reduced from $P_{NF}$ to $P_F$, even without large changes in quantities consumed ($Q_{NF}$-$Q_F$).

These simple diagrams give an indication of the forces at play in dairy markets because of fasting. The expected downward pressure on prices in butter markets is partly explained by opportunity costs for the storage of butter and should in more complete models be addressed by more sophisticated models that incorporate the costs of processing and storage. Other adaptation methods and processing possibilities for milk should be considered as well. However, these considerations are beyond the scope of this analysis.

The purpose of this simple conceptual framework is to help set the stage for expected results of our analysis. First, we expect that dairy consumption by those not affected by fasting will go up because of price decreases. Second, milk production will go down during (longer) fasts. Third, price swings will be larger for storable dairy commodities than for liquid milk as storable products will be affected by supply increases on top of demand decreases. Finally, the predicted effects all depend on the magnitude of fasting adherence by the population at large and on the level of market integration, as they determine the size of these demand shifts.

## Data and empirical strategy

We use unique primary data collected within the project 'Improving the evidence and policies for better performing livestock systems in Ethiopia' led by the International Food Policy Research Institute as part of the Feed the Future Innovation Lab for Livestock Systems with the aim of mapping the supply chain of dairy products to Ethiopia's capital city, Addis Ababa, the most developed milk shed within the country. Data collection was focused on the zones of North and West Shewa as dairy production in these zones is the highest of all surrounding zones of Addis Ababa. We surveyed 870 milk producing households with at least one cow in milk located in and around Addis Ababa (30 urban, 240 suburban, and 600 rural producers).

Data collection took place between 22 January and 15 February 2018. The Institutional Review Board of the International Food Policy Research Institute ethical granted approval for this study. The survey was carried out in collaboration with the Ethiopian Development Research Institute, a think tank linked to the government of Ethiopia. Consent was informed and written. The data are publicly available on the Harvard Dataverse website see: https://dataverse.harvard.edu/dataset.xhtml?persistentId=doi:10.7910/DVN/7AIKZH. We focused on those households with no more than 20 cows.

This paper looks at three research questions, as indicated in the introduction. It is important to keep in mind that 85 percent of the surveyed households reported being Orthodox Christian. The other surveyed households reported being Protestant (12%), Muslim (2%), Catholic (1%) or indicated that they adhere to other religious beliefs, e.g., traditional faiths (1%). To answer our first research question on effective fasting participation, we only examine data obtained from Orthodox households in our sample. For the remaining research questions on production adaptation and consumption, our analytical sample is made up of both Orthodox and non-Orthodox households. In our regressions for these research questions, we include controls for religious affiliation of the household.

Our first study question is to assess the effective number of days annually that members of Orthodox households fast. To do so, we asked respondents (usually the household head) in Orthodox households to indicate for each household member above the age of five years whether or not this member observes any Orthodox fasting event. If a member was said to engage in fasting, we detailed for each of the seven major fasts, as outlined in Table 1, a member's observance with a dummy variable (yes/no). If a member participates in a particular fasting event, we assume that this member fasts during the entire prescribed fasting period. This method provides us with fasting data on 3,946 individuals. We extend this analysis by investigating the factors associated with fasting participation. To do so, we run a random and household-level fixed effect logit regression.

Second, to detail the impact of fasting on the value chain, we look both at the adaptation strategies of milk producers and at dairy prices. We start by analyzing the different strategies used by farmers to deal with fasting by looking at the impact fasting has on their milk production and the different milk output uses they adopt.

We further explore potential heterogeneity in adopted adaptation strategies, specifically that dependent upon the household's degree of market access. We proxy market accessibility by remoteness to Addis Ababa, assuming that households located closer to the capital have better market access. The variable 'Remoteness to Addis Ababa' is defined and calculated in line with Vandercasteelen, Beyene [35] using the Ethiopian road network and quality data (obtained from the Ethiopian Road Authority) combined with the farmer's self-reported travel time to the nearest road segment. The obtained measure thus captures the travel time (expressed in hours) from the farm's location to Ethiopia's capital city. We categorize farmers in two groups, remote and non-remote, using the median of 1.54 hours travel time to Addis Ababa as a cut-off between the two groups.

To appraise price effects linked with fasting, we use market prices reported by our participants over the 12 months preceding the survey. We further detail the effect of remoteness on seasonal price swings caused by fasting. We do so by using the reported prices that farmers obtained for dairy products that they sold in the middle of the most recent long fasting season and two weeks after this fasting season, when they had cows in milk. To assess these impacts of fasting, we run fixed effect regression models at the household level, splitting the data between remote and non-remote milk producing households, following the definition mentioned above.

Finally, we assess the impact of fasting on dairy intake of the household as a whole and the youngest child aged five years and under separately. Furthermore, we test with detailed data

what was consumed by the youngest child in the day before the survey and how this is affected by the fasting observance of the household head. Fasting observance is self-reported and equaled one if the household head indicated to have fasted the day before the survey. We again look at heterogeneity in these effects linked to market access.

To assess the impact of fasting on producer adaptation strategies and consumption, we use Ordinary Least Squares (OLS) and Tobit regressions. Using the time variation of our interview period (Table 2), we were able to develop two fasting indicators for each household surveyed, based on the number of consecutive official fasting days in the week and in the month prior to the interview. These fasting indicators are calculated using the official fasting calendar (as reflected in Table 1). We calculate these indicators on a weekly and monthly basis to accommodate the fact that outcome variables are based on either weekly (consumption) or monthly (production and output use) recall periods.

For both fasting indicators, a cut-off is chosen to distinguish between short and long fasting episodes. The cut-off is set at the average amount of consecutive fasting days, which for the weekly and monthly fasting indicator amounts to 4 days (varying between 1 and 7 days) and 10 days (varying between 3 and 17 days), respectively (Table 3). By dividing the sample in this way, we construct a treatment variable where the treatment equates the presence of a long fasting period in the past week or past month. In our sample, 31 and 60 percent of households were confronted with a long fasting period in the last week and month, respectively. The difference in likelihood of treatment between both fasting indicators can be explained by the presence of the Advent fasting period, which is covered in the monthly fasting indicator, but not in the weekly indicator.

Besides these main explanatory variables, we also included household and individual variables outlined in S1 Appendix as controls. Regressing the fasting indicators on all covariates, using an F-test, revealed that neither of the two treatment variables withstands a joint orthogonality test. It shows, for example, that the variation in the sample across space (remoteness) is related to the variation across time (fasting indicators) in a non-random way. The controls include household socio-demographic characteristics, as well as controls related to dairy production and marketing. We also control for within-sample concentrations of different religions in the village where a household resides. We assume that religious diversity in the immediate vicinity of a household, and particularly the concentration of Orthodox followers, may impact milk production and milk output use decisions. Finally, we include a series of individual control variables specifically related to the milk intake of children. In all regressions, standard errors are clustered at the village level.

Descriptive statistics of the dependent variables used in the OLS and Tobit regressions are summarized in Table 4. Average daily milk production in our sample is 3 liters per cow, of which 55 percent is processed into cheese, butter, or buttermilk, 25 percent is sold, 19 percent is used for own consumption, and a minor share (0.5%) is given away. Sales volumes vary considerably for all dairy products, driven in part by the fact that an important share (35%) of the dairy producers does not sell any dairy products. Still, it can be observed that producers specialize in liquid milk sales, with processing at the farm level mainly intended for own

**Table 2. Timing of fasting periods and data collection, 2017 and 2018.**

| Dates | 28 Nov to | 07 Jan to | 22 Jan to | 29 Jan to | 01 Feb to | 12 Feb to | 26 Feb to |
|---|---|---|---|---|---|---|---|
| | 06 Jan | 21 Jan | 28 Jan | 31 Jan | 11 Feb | 25 Feb | 07 Apr |
| Fasting period | Advent | Weekly only | Weekly only | Weekly only | Nineveh | Weekly only | Weekly only |
| Data collection | | | | | | | |

**Table 3. Descriptive statistics of the constructed fasting indicators.**

| Variable | Unit | Mean (standard deviation) |
|---|---|---|
| Consecutive fasting days week | number (1–7) | 3.09 (2.48) |
| **Weekly fasting indicator** | 0/1 | 0.31 |
| Consecutive fasting days month | number (3–17) | 9.44 (3.71) |
| **Monthly fasting indicator** | 0/1 | 0.60 |
| Observations | | 870 |

consumption. Dairy products consumed within the household are almost exclusively sourced from own production (99.8% on average across all dairy products).

We collected separate milk intake data for the youngest child below the age of five years in each surveyed household. Average household milk intake fluctuates around 3 liters, of which the youngest child consumes on average 2 liters. Our data thus confirm that young children are prioritized when it comes to milk consumption, in line with the prevailing assumption in Ethiopia that milk is mainly meant for children [36]. Wastage of milk and related products does not appear to be an issue in our sample. A study by Minten, Tamru [37], using the same producer data as this paper, assessed that the percentage of all produced milk spoiled at farm level was as little as 0.003 to 0.008 percent.

## Results

### Fasting participation and beliefs

Table 5 summarizes the extent of participation in fasting periods throughout the year for Orthodox households. The number of fasting days on a yearly basis varies from 5.2 days for

**Table 4. Descriptive statistics of outcome variables used in analysis of milk producer adaptation strategies and consumption.**

| Variable | Unit | Household | Youngest child |
|---|---|---|---|
| **Past month** | | | |
| Milk production | l/cow/day | 2.95 (3.85) | |
| Share milk home processed | % | 54.98 (37.83) | |
| Share milk sold | % | 25.62 (40.31) | |
| Share milk home consumed | % | 18.78 (21.74) | |
| Share milk given out | % | 0.50 (3.59) | |
| Sales quantities past month | | | |
| *Milk* | l | 123.9 (351.7) | |
| *Butter* | kg | 1.08 (2.87) | |
| *Cheese* | kg | 7.90 (31.18) | |
| **Past week** | | | |
| Dairy consumption | | | |
| *Milk* | l | 3.16 (4.44) | 2.04 (2.67) |
| *Buttermilk* | l | 1.70 (3.30) | 0.40 (1.09) |
| *Butter* | l eq.[a] | 2.16 (5.79) | |
| *Cheese* | l eq. | 1.64 (3.57) | |
| Observations | | 870 | 317 |

*Note*. Means are shown with standard deviations in parentheses.

[a] For comparison purposes, kilogram (kg) amounts of processed products are transformed into liter (l) of milk equivalents. The conversion rates used are 20 l/kg for butter and 1.33 l/kg for cheese.

**Table 5. Participation rates and fasting days per age group for members of the Orthodox faith.**

|  | Participate in any fasting | Number of fasting days | |
|---|---|---|---|
|  | % | Average | Standard deviation |
| Young children (5–6) | 4.97 | 5.19 | 29.52 |
| Older children (7–9) | 22.75 | 34.08 | 65.97 |
| Adolescents (10–19) | 64.51 | 98.40 | 77.63 |
| Adults (20–64) | 88.53 | 140.36 | 57.91 |
| Elderly (65+) | 89.42 | 141.65 | 56.86 |

young children to 142 days for elderly people. The fasting adherence of adults is around 140 days and thus 26 to 110 days below the estimates of 166 to 250 fasting days per year reported elsewhere [11, 16, 17]. Both observations could indicate that fasting practices are changing over time and that younger people–who are increasingly exposed to more liberal and thus less strict fasting practices–become less adherent. It is also to be noted that between 11 and 12 percent of Orthodox adults do not participate in any fast, possibly because of pregnancy or illness, but possibly out of choice. Furthermore, fasting adherence increases with age. Average participation among older children, who are supposed to participate in fasting periods, is rather low with about one-in-five participating. Adolescents fast more commonly, 65 percent on average, while observance reaches 90 percent among adults and the elderly.

Only a small proportion of young children participates in fasting rituals, although they are in principle exempt from fasting. Among Orthodox surveyed households, we asked opinions about fasting observance by young children. Only 11 percent of the households disagreed or strongly disagreed that children less than 2 years old should not observe fasting. Four percent believed that children aged 6 to 23 months should stop eating ASF during fasting; 6 percent argued that these children should eat less frequently during fasting; and 5 percent though these children should be breastfed less during fasting. In general, Orthodox households in our sample stated that children should start observing fasting at the age of 9.2 years (with a 95 percent confidence interval of 8.6 to 9.7 years old). While these findings are lower than those reported by Kim et al. [30], who find that only half of caregivers had positive attitudes towards the exemption of children from fasting practices, it is still worrisome to find in our data that a number of young children are actively participating or are expected to participate in fasting.

To understand factors associated with fasting participation, we run a random and household-level fixed effect logit regression. A number of interesting results show up from that exercise (Table 6). First, participation rates differ greatly across the different fasting events. We observe that Lent, Felseta fast, and the weekly fasting days are the most adhered to, followed, at some distance, by the Nineveh and the Advent fasts. Second, women participate significantly less in fasting events than men. Third, as noted earlier, we also see strong associations with age. Finally, we note that household head controls, household size, and household income have no significant association with fasting participation. We do however observe significantly more fasting participation the farther that the household is located from Addis Ababa, whereas an increased concentration of Orthodox households at the village level significantly reduces fasting participation.

The results indicate that fasting is widespread in the Orthodox community, but that there is also significant heterogeneity in fasting adherence. Using these data, we calculate that annual milk consumption at the national level is reduced by about 12 percent because of Orthodox fasting practices–this calculation is based on 43 percent of the population being Orthodox (CSA 2010); 38 percent of all days being fasting days (140 days out of 365); and children below 10 years of age, who represent 28 percent of the Orthodox population, being exempt from

**Table 6. Factors associated with fasting participation.**

|  | Unit | Random effect | | Fixed effect | |
|---|---|---|---|---|---|
|  |  | Coefficient (S.E.) | z-value | Coefficient (S.E.) | z-value |
| **Fasting events** |  |  |  |  |  |
| Advent fast | 0/1 | -4.64 (0.20) | -23.76 | -4.36 (0.19) | -22.90 |
| Lent fast | 0/1 | -1.27 (0.21) | -6.03 | -1.21 (0.21) | -5.90 |
| Nineveh fast | 0/1 | -4.14 (0.19) | -21.28 | -3.90 (0.19) | -20.53 |
| Sene fast | 0/1 | -6.04 (0.20) | -30.29 | -5.69 (0.19) | -29.29 |
| Felseta fast | 0/1 | -1.27 (0.21) | -6.03 | -1.32 (0.20) | -6.45 |
| Epiphany fast | 0/1 | -7.61 (0.21) | -36.19 | -7.41 (0.21) | -35.80 |
| Weekly fasting | 0/1 | Omitted |  | Omitted |  |
| **Individual controls** |  |  |  |  |  |
| Sex: female | 0/1 | -0.16 (0.06) | -2.55 | -0.14 (0.06) | -2.31 |
| Age |  |  |  |  |  |
| 5 to 6 years | 0/1 | -3.97 (1.14) | -3.47 | -4.96 (1.14) | -4.34 |
| 7 to 9 | 0/1 | 2.48 (0.93) | 2.67 | 2.24 (0.91) | 2.47 |
| 10 to 19 | 0/1 | 2.86 (0.92) | 3.11 | 2.64 (0.89) | 2.97 |
| 20 to 64 | 0/1 | 3.28 (0.92) | 3.57 | 3.06 (0.89) | 3.44 |
| 65+ | 0/1 | 3.59 (0.93) | 3.88 | 3.38 (0.90) | 3.76 |
| Schooling | years | 0.01 (0.01) | 0.55 | 0.00 (0.01) | 0.38 |
| **Household head controls** |  |  |  |  |  |
| Sex: female | 0/1 | -0.57 (0.55) | -1.04 |  |  |
| Age | years | 0.02 (0.01) | 1.51 |  |  |
| Schooling | years | 0.04 (0.04) | 1.18 |  |  |
| **Household controls** |  |  |  |  |  |
| Household size | # | 0.04 (0.07) | 0.57 |  |  |
| Total income | '000 ETB/month | -0.01 (0.03) | -0.19 |  |  |
| Orthodox concentration | % | -3.19 (0.97) | -3.29 |  |  |
| Remoteness | hours | 0.25 (0.11) | 2.19 |  |  |
| Observations |  | 19,306 | | 12,642 | |
| Chi-squared |  | 2,915.26 | | 6,986.12 | |

fasting. While the effects of fasting on milk consumption in specific periods and locations might obviously be higher, this result suggests that fasting is only a partial explanation for low dairy consumption in the country. We now look at milk producer adaptation strategies, pricing effects, and consumption changes because of this fasting practice.

## Fasting, adaptation strategies, and dairy prices

**Fasting and producer adaptation strategies.** In this section, we look at different adaptation strategies developed by milk producing households (both Orthodox and non-Orthodox) to manage the reduced demand for liquid milk during fasting periods. We first look at production and output use. We find that long fasting periods reduce daily milk production by one-fourth on average (Table 7). This suggests that farmers are aligning their production with fasting periods, especially since part of the data collection period coincided with the Advent fasting, a period during which milk output should be at its highest level [19]. This confirms the predicted effect from the conceptual model.

Furthermore, we observe that long fasting seasons reduce marketing possibilities, with especially milk sales dropping significantly (Table 7). Farmers turn the milk that is not sold into

**Table 7. Effect of the presence of a long fasting season in the last month on milk production and output use one month prior to the interview.**

| | Milk production and output use last month | | | | |
|---|---|---|---|---|---|
| | **Milk production (l/day)** | **Milk sold (%)** | **Milk processed (%)** | **Milk consumed (%)** | **Milk given out (%)** |
| Fasting past month (0/1) | -1.67* | -48.71*** | 19.64*** | -1.15 | -8.05 |
| | (0.97) | (14.22) | (6.90) | (2.18) | (8.84) |
| Model employed | OLS | Tobit | Tobit | Tobit | Tobit |
| Household controls | Yes | Yes | Yes | Yes | Yes |
| Non-fasting mean | 7.03 | 40.44 | 41.66 | 17.40 | 0.50 |
| Observations | 855 | 855 | 855 | 855 | 855 |
| (Pseudo) R-squared | 0.72 | 0.19 | 0.07 | 0.02 | 0.09 |

*Note*. See S1 Appendix for a complete list of controls. Total dairy production is not included as a control variable. Cluster robust standard errors in parentheses.

*** $p < 0.01$,

** $p < 0.05$,

* $p < 0.1$

dairy products with a longer shelf life, processing up to 60 percent of their milk output during long fasts. The share of consumption does not change with the presence of long fasting periods, which translates into lower absolute levels of consumption given the decrease in milk production (for more detail, see Table 11). We further observe no change in the already low share of milk being given away. As we have no reliable estimates of wastage of milk, it is possible that a large part of the remaining milk is thrown away during fasting periods.

Regarding sales volumes, we find that only milk sales are significantly affected by long fasting periods, being reduced on average by 28 percent (Table 8). Milk sales do not completely drop to zero, since some milk buyers continue purchasing milk during Orthodox fasting. This is the case for about 18 percent of the farmers in our sample. Anecdotal evidence suggests that some buyers keep on buying milk, although they do not need nor use this milk but rather throw it away afterwards [38]. Continuing buying milk during Orthodox fasting seasons

**Table 8. Effect of the presence of a long fasting season in the last month on total dairy sales one month prior to the interview.**

| | Total monthly dairy sales last month | | |
|---|---|---|---|
| | **Milk (l)** | **Butter (kg)** | **Cheese (kg)** |
| Fasting past month (0/1) | -48.86* | 0.59 | 4.00 |
| | (24.89) | (0.35) | (3.38) |
| Model employed | OLS | OLS | OLS |
| Household controls | Yes | Yes | Yes |
| Non-fasting mean | 173.35 | 0.70 | 5.37 |
| Observations | 855 | 855 | 855 |
| R-squared | 0.77 | 0.25 | 0.26 |

*Note*. See S1 Appendix for a complete list of controls. Cluster robust standard errors in parentheses.

*** $p < 0.01$,

** $p < 0.05$,

* $p < 0.1$

seemingly helps buyers to assure supply from dairy producers in the non-fasting periods. Using a robustness check (S3 Appendix), we find that long fasting periods affect the decision to sell milk, but not sales volumes conditional on a farmer selling milk during a fast.

We further assess heterogeneity of adaptation strategies with respect to remoteness (Table 9). We find that remote households have predominantly adapted to a culture of fasting by reducing their milk production, with total milk output falling by 60 percent during long fasting periods. Non-remote households, on the other hand, seem to have better access to buyers who continue purchasing milk during fasting, which enables them to maintain their milk production. However, we observe a slightly negative, yet not significant, trend in total milk output in these non-remote households as well. But, we find considerable heterogeneity in their ability to sell milk during fasting seasons and overall it seems that milk sales volumes tend to go down slightly. Milk sales of remote farmers, on the other hand, drop significantly (80%). As a result, both remote and non-remote households rely on processing to overcome Orthodox fasting events. For neither remote nor non-remote households, however, do sales of processed milk products increase during long fasting periods.

**Fasting and dairy prices.**   Looking at price effects linked with fasting, we find that prices for different dairy products are low during periods of abundant supply (from June until the end of the year) and low demand (the Advent and Lent fasting periods). Conversely, prices are at their highest level when high demand, such as for feasts like Christmas and Easter, interacts with a declining availability of dairy products, such as in January and February (see S2 Appendix). Fig 2 also shows that price fluctuations are most pronounced for processed milk products, while milk prices seem to be barely affected by seasonality in supply or demand, although milk prices are slightly lower during Lent. Similar seasonal price variations for dairy products are reported by Bachewe, Minten [16]. These results therefore confirm the predicted effect from the conceptual model.

We further look at the effect of remoteness of farmers on seasonal price swings (Table 10). We continue to see with these data lower price swings during the fasting seasons for liquid milk compared to processed milk products. While fasting leads to a price reduction of 3 to 5 percent for liquid milk, this is as high as 7 to 8 percent for butter, and 18 percent for cheese.

**Table 9. Heterogeneity in milk production and output use adaptation strategies by remoteness to Addis Ababa.**

| | Milk production and output use last month | | | | | | | |
|---|---|---|---|---|---|---|---|---|
| | Milk production (l/day) | | Processing (%) | | Milk sales (l) | | Processed sales (kg) | |
| | *Remote* | *Not remote* | *Remote* | *Not remote* | *Remote* | *Not remote* | *Remote* | *Not remote* |
| Fasting past month (0/1) | -1.51** | -1.59 | 11.14 | 27.00*** | -36.86** | -40.34 | 3.60 | 3.49 |
| | (0.68) | (1.41) | (7.02) | (8.47) | (17.30) | (31.58) | (4.80) | (2.92) |
| Model employed | OLS | OLS | Tobit | Tobit | OLS | OLS | OLS | OLS |
| Household controls | Yes | Yes | Yes | Yes | Yes | Yes | Yes | Yes |
| Non-fasting mean | 2.32 | 10.42 | 63.30 | 26.06 | 46.12 | 265.14 | 7.41 | 5.10 |
| Observations | 427 | 428 | 427 | 428 | 427 | 428 | 427 | 428 |
| (Pseudo) R-squared | 0.39 | 0.77 | 0.02 | 0.08 | 0.30 | 0.86 | 0.60 | 0.20 |

*Note.* Remote / Not remote = Below / Above median travel time to Addis Ababa (1.54 hours). See S1 Appendix for a complete list of controls. Remoteness and access to the Addis market are not included as control variables and neither is total production for the regressions where milk production and share of processing are the dependent variables. Cluster robust standard errors in parentheses.

*** p<0.01,

** p<0.05,

* p<0.10

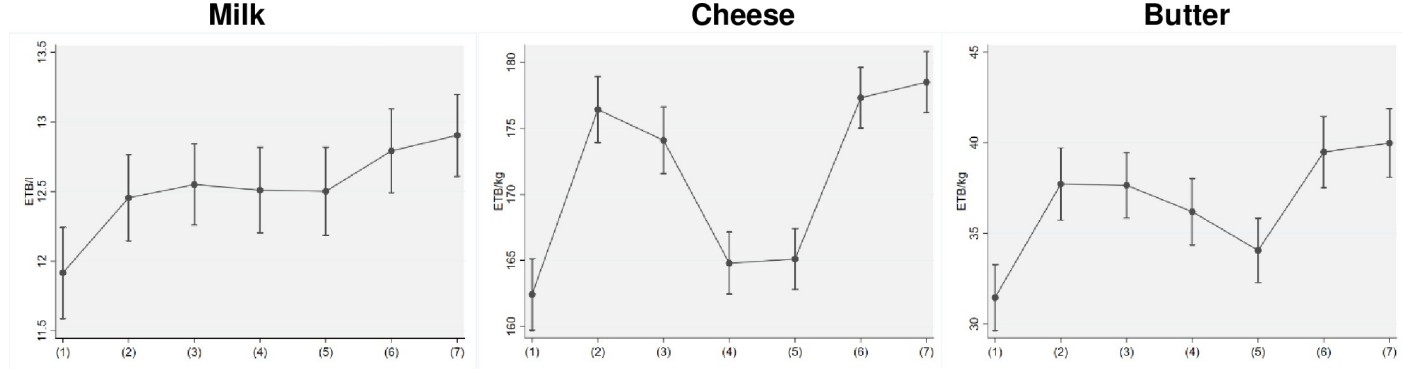

(1) Lent; (2) Easter; (3) Easter-May; (4) Jun-Advent; (5) Advent; (6) Christmas; (7) Christmas-current
*Note*: ETB = Ethiopian Birr. At the time of the survey: 1 USD = 27 ETB. Whisker-plots around points
are 95-percent confidence intervals.

**Fig 2. Average market prices of different dairy products throughout the year.**

We also note that for milk there are higher price differences for remote areas compared to non-remote areas. This indicates the importance of access to markets for highly perishable milk to reduce price volatility because of fasting. The limited effect of fasting on liquid milk prices seems to suggest that the magnitude of demand and supply shifts caused by fasting is almost equal for liquid milk, at least based on our data.

## Fasting and consumption

Finally, we take a closer look at the impact of fasting on dairy consumption with detailed data at household level and for children below the age of five years. If there was a long fasting event in the week prior to the interview, households' dairy consumption decreases, but only significantly so in the case of butter, with intake almost cut by half (Table 11). Such drops are expected, since the majority of producing households (85%) are Orthodox Christian (S1 Appendix). The limited impact of fasting on weekly dairy consumption could indicate that adult household members compensate for the forgone consumption of dairy, as well as other

**Table 10. Fixed effects analysis of heterogeneity in price differences by remoteness to Addis Ababa.**

| | Dairy prices | | | | | |
|---|---|---|---|---|---|---|
| | Milk (ETB/liter) | | Butter (ETB/kg) | | Cheese (ETB/kg) | |
| | *Remote* | *Not remote* | *Remote* | *Not remote* | *Remote* | *Not remote* |
| Fasting (0/1) | -0.559*** | -0.396*** | -12.24*** | -13.69*** | -5.94*** | -8.09*** |
| | (0.119) | (0.099) | (1.617) | (1.401) | (0.579) | (0.996) |
| Model employed | FE | FE | FE | FE | FE | FE |
| Non-fasting mean | 10.96 | 13.37 | 168.42 | 175.70 | 32.49 | 44.58 |
| Observations | 60 | 234 | 305 | 183 | 118 | 151 |

Note. Remote / not remote = below / above median travel time to Addis Ababa (1.54 hours).

*** p<0.01,

** p<0.05,

* p<0.1

**Table 11. Effect of the presence of a fasting period of more than four days in the last week on dairy consumption one week prior to the interview.**

| | Dairy consumption past week | | | | |
|---|---|---|---|---|---|
| | **Milk** | **Butter** | **Cheese** | **Buttermilk household** | **Milk youngest child** |
| | household | household | household | | |
| | (l/ad.eq.) | (l/ad.eq.) | (l/ad.eq.) | (l/ad.eq.) | (l) |
| Fasting past week (0/1) | -0.14 | -0.19** | -0.13 | -0.049 | 1.02*** |
| | (0.10) | (0.08) | (0.092) | (0.055) | (0.33) |
| Model employed | OLS | OLS | OLS | OLS | OLS |
| Individual controls | No | No | No | No | Yes |
| Household controls | Yes | Yes | Yes | Yes | Yes |
| Non-fasting mean | 0.64 | 0.41 | 0.34 | 0.43 | 1.76 |
| Observations | 848 | 849 | 841 | 847 | 267 |
| R-squared | 0.16 | 0.12 | 0.11 | 0.31 | 0.22 |

*Note*. Household amounts are expressed per adult equivalent to account for household size and composition. For complete list of controls, see S1 Appendix. Cluster robust standard errors in parentheses.

*** p<0.01,

** p<0.05,

* p<0.1

ASF, during non-fasting days. Strikingly, it appears that households' milk consumption during long fasting episodes would be even lower, were it not for the fact that the youngest children are given considerably more milk in case of a long fasting event in the past week. Their milk consumption increases by 60 percent during long fasting periods–they consume on average one extra liter of milk. This is in line with the predictions in the conceptual model.

For children aged five years and below, we test with detailed data what was consumed in the day before the survey (Fig 3). Most of the children consume milk (72 percent) while the consumption of pasteurized milk and powder milk is limited (2 percent). This is not surprising given the availability of fresh milk in these producing households. Children do consume cheese and butter, although the likelihood is much lower than compared to milk (20 to 25 percent). The probability of consuming other ASF, such as meat, fish, or yoghurt, is below 10 percent, except for eggs at around 50 percent. We find that a single fasting day does not impact negatively on their overall intake of dairy products. If anything, we see an increase in consumption of milk by children during fasting days, although this increase is not significant at conventional statistical levels. But, we observe that children consume less beef, pork, and lamb on fasting days, probably because these food items are less likely to be consumed and prepared at the household level during fasts.

One of the strategies that households have developed to overcome long Orthodox fasting periods is, thus, to allocate part of the surplus milk to their young children. These findings are different from those reported by Kim et al. [30] who observed that, even in a setting where 80 percent of the interviewed households own livestock, only 25 percent of children consume any ASF during the Lent fasting period. An important reason for the increased consumption of milk by children that we observe seems to be the increased transaction costs to market milk during fasts. Marketing opportunities for milk drop significantly during fasting (Table 7) with buyers purchasing less or no milk during these periods. This increases transaction costs dramatically if farmers would want to search for alternative buyers to whom they could sell their milk. Since market prices of milk decrease during fasting (Table 9), a preferred option for farmers might be to channel part of the non-sold milk to their children. Of course, there is a limit to how much milk young children can consume. In our data, there appears to be an

**Solid food**

**Liquid foods**

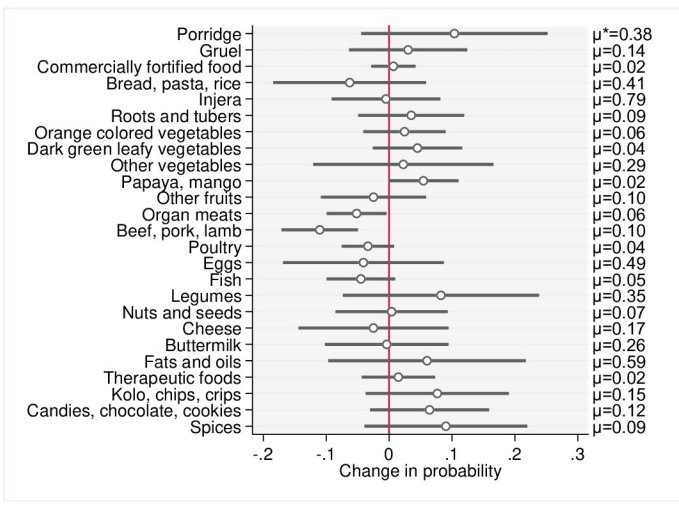

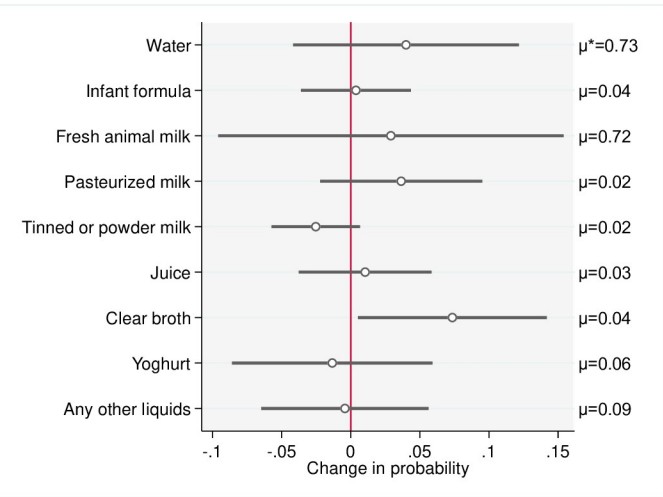

*Note.* *μ denotes the non-fasting mean of the dependent variable. Coefficients of the fasting day indicator are shown along with cluster robust 95-percent confidence intervals. For complete list of controls, see Supporting information S1.

**Fig 3. Impact of a fasting day on the likelihood of daily consumption of different food groups for the youngest child.**

upper limit of about 7 liters per week for the youngest child. This implies that even during long fasting periods, allocating more milk to children can only partly solve the excess milk supply problem.

We also test heterogeneity in consumption with respect to remoteness (Table 12). There is large variability in milk consumption patterns among the non-remote households with some households consuming more and others consuming less milk during fasting periods. Remote households on the other hand, significantly reduce the amount of milk and processed milk

**Table 12. Heterogeneity in consumption adaptation strategies by remoteness to Addis Ababa.**

| | Dairy consumption last week | | | | | |
| --- | --- | --- | --- | --- | --- | --- |
| | Milk production (l/day) | | Processed household (l/ad.eq.) | | Milk child (l) | |
| | *Remote* | *Not remote* | *Remote* | *Not remote* | *Remote* | *Not remote* |
| Fasting past week (0/1) | -0.25** | -0.02 | -0.64*** | -0.24 | 0.98 | 1.45*** |
| | (0.09) | (0.12) | (0.16) | (0.20) | (0.86) | (0.36) |
| Model employed | OLS | OLS | OLS | OLS | OLS | OLS |
| Household controls | Yes | Yes | Yes | Yes | Yes | Yes |
| Child controls | No | No | No | No | Yes | Yes |
| Non-fasting mean | 0.57 | 0.78 | 1.33 | 0.85 | 1.59 | 2.11 |
| Observations | 422 | 426 | 420 | 419 | 133 | 134 |
| R-squared | 0.21 | 0.19 | 0.24 | 0.18 | 0.27 | 0.34 |

*Note.* Remote / not remote = below / above median travel time to Addis Ababa (1.54 hours). For a complete list of controls, see S1 Appendix. Remoteness and access to the Addis market are not included as control variables. Cluster robust standard errors are in parentheses.

*** p<0.01,

** p<0.05,

* p<0.10.

products consumed during fasting. Yet, both types of households allocate more milk to children during fasting, significantly so among non-remote households.

Finally, when comparing daily intake of children below the age of five years in remote and non-remote households, we find that fewer children consume fresh milk and buttermilk in non-remote households (64 percent of children in non-remote households versus 75 percent in remote households and 9 percent versus 35 percent, respectively). The consumption of pasteurized milk, powdered milk, and cheese does not vary between remote and non-remote households. Overall, we observe that a single fasting day seems to positively impact the intake of milk in non-remote households (as was observed in Table 12), whereas the opposite is observed for consumption of pasteurized milk (Fig 4). These figures suggest that non-remote households prioritize milk sales during non-fasting periods, and thus allocate less milk to their children, whereas surplus milk gets increasingly allocated to children in these households during fasting periods. To provide further background, similar figures for the household head and the spouse are included in S4 Appendix.

## Discussion and conclusion

The impact of food taboos on food consumption and value chains is not well understood. Fasting practices embedded within the largest religious community of Ethiopia, Orthodox Christians, create significant variation in demand for dairy products. However, evidence on adherence to fasting and its implications for the adaptation strategies of milk producers, for the pricing of dairy products, and on dairy consumption is limited. We study these issues in this paper for dairy producers. Choosing this particular target group allows us to measure the simultaneous impact of fasting on dairy intake within the producer households, as well as on production, processing, and marketing decisions of dairy producers. No such comprehensive study has been undertaken in the Ethiopian context before.

We find that fasting adherence is widespread and that the average Orthodox Christian adult fasts for 140 days a year. Milk producing households adopt diverse strategies to overcome periods of reduced milk demand associated with fasting. Overall, we observe that such households are affected by two impact pathways during fasting periods. First, these households lower their own intake of dairy during fasting, which creates a surplus of milk at the household level. Second, as market opportunities are reduced during fasts, there are fewer outlets available for the surplus liquid milk. To manage this surplus milk, farmers adopt a combination of three strategies: (1) they reduce total milk output by aligning the number of cows in milk with fasting periods, (2) they increasingly channel produced milk to their youngest children, and (3) they expand processing of milk into less perishable dairy products, such as cheese or butter. The importance of each of these strategies, however, varies significantly along with the degree of access a milk producing household has to the major market of Addis Ababa. Remote households reduce their milk production significantly during longer fasting periods, process some of it into cheese or other dairy products, or feed some of their excess milk production to their children. Non-remote households, on the other hand, have been able to establish better arrangements with milk buyers so that they do not need to stop producing milk during fasting periods. Yet, during fasts, these milk producing households channel significantly more milk to their children and also process more into less perishable products. Furthermore, we find differential effects on prices for different dairy products, with small seasonal swings for liquid milk prices and larger ones for processed milk products.

Our results have several important policy implications. First, as the impact of fasting on milk consumption at national level is found to be relatively small and we find that children consume milk when it is available even during fasting periods, this suggests other issues such

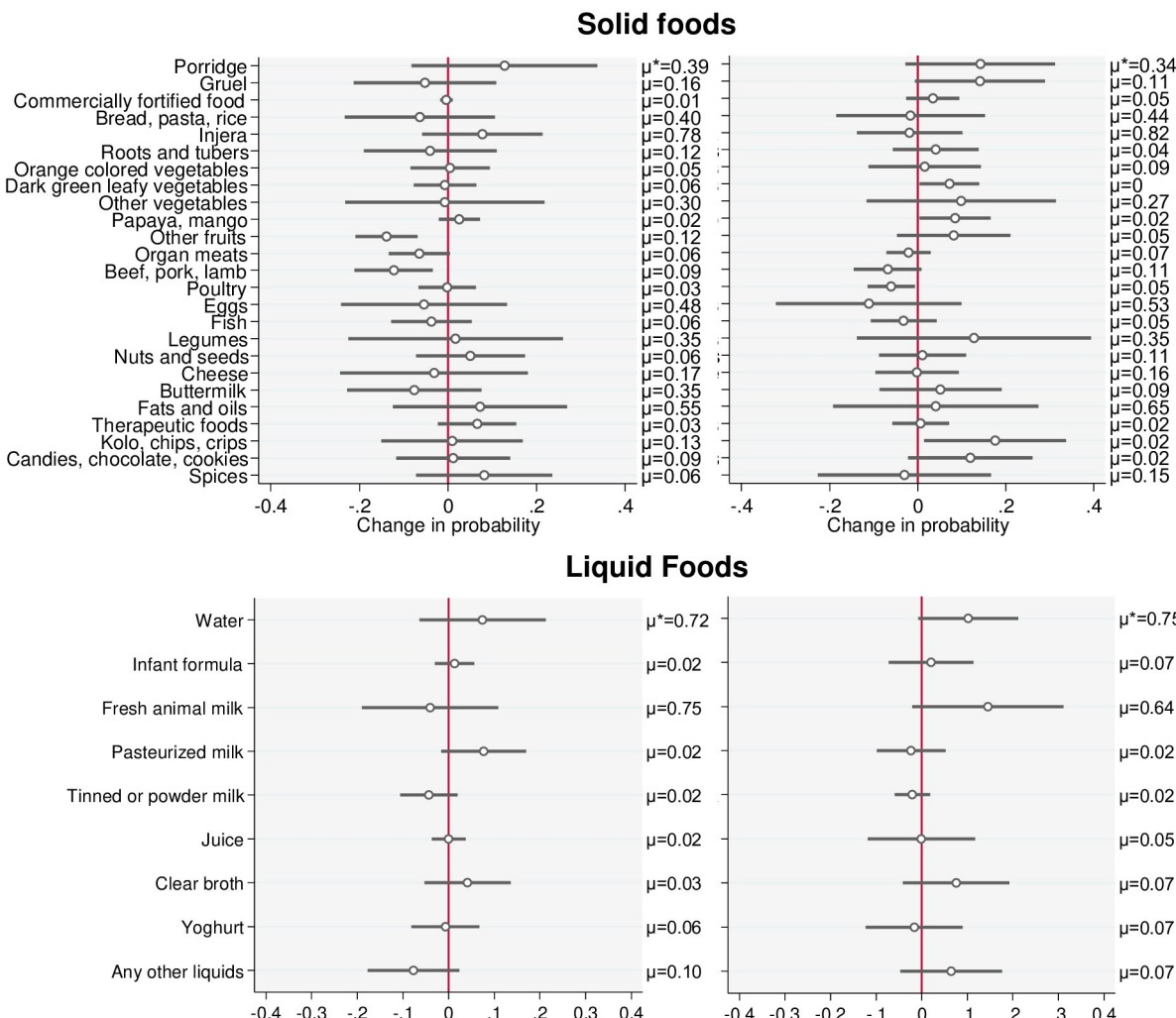

**Fig 4. Impact of a fasting day on likelihood of daily consumption of different food groups for children in remote and non-remote households.**

as availability and affordability–instead of fasting–are the main impediments to increased dairy consumption in Ethiopia. Further investments to stimulate the dairy sector are therefore needed to increase availability of dairy products at lower prices. In contrast to the rapid growth in crop output and productivity recorded in Ethiopia, ASF output has grown slowly and productivity has stagnated, seemingly due to low availability and adoption of improved inputs in the sector. The result has been high and increasing prices for dairy products [39].

Second, we see a (small) number of children participating in fasts, even though they are, in principle, exempt from fasting. Further efforts at widely communicating the potential adverse developmental effects of fasting on children is needed.

Finally, to help smoothen the effects of demand swings for milk and other dairy products and possibly to increase returns to investments in the sector, further efforts are needed towards enhancing processing practices (such as increasing production of ultra-heat treated and powdered milk), ensuring availability of milk chilling centers, and improving transportation facilities in milk sheds and regionally to assure market integration and to expand dairy marketing to areas where fasting is less prevalent.

Furthermore, our findings point to a number of areas for further research. First, since our sample is not representative of the population at large, it would be useful to collect similar data with the same level of detail among all households, not only dairy producers. Second, future research would benefit from more focus on health outcomes and from repeated observations over time. Third, studies, in Ethiopia as well as in other countries, could target different religious groups to assess how their food habits and beliefs are related to food consumption and the development of food value chains. Finally, only dairy has been looked at in this study. It would be good to broaden the study to other ASF.

## Supporting information

**S1 Appendix. Control variables.**
(DOCX)

**S2 Appendix. Seasonal milk production.**
(DOCX)

**S3 Appendix. Market participation and supply.**
(DOCX)

**S4 Appendix. Daily consumption of different food groups–household head and spouse.**
(DOCX)

## Acknowledgments

Most of this article was written before Eline D'Haene joined the Department of Plants & Crops, before Senne Vandevelde joined the European Commission and before Bart Minten joined IFPRI-Yangon. Opinions expressed in this article are those of the authors and do not necessarily reflect the view of their institutions. We would also like to thank seminar participants at LICOS, KU Leuven and at the University of Ghent for their constructive feedback. We are especially grateful to Seneshaw Tamru Beyene for excellent field support.

## Author Contributions

**Conceptualization:** Eline D'Haene, Senne Vandevelde, Bart Minten.

**Data curation:** Senne Vandevelde, Bart Minten.

**Formal analysis:** Eline D'Haene, Senne Vandevelde.

**Funding acquisition:** Bart Minten.

**Investigation:** Senne Vandevelde, Bart Minten.

**Methodology:** Eline D'Haene, Senne Vandevelde, Bart Minten.

**Supervision:** Bart Minten.

**Validation:** Eline D'Haene, Senne Vandevelde, Bart Minten.

**Visualization:** Eline D'Haene, Senne Vandevelde, Bart Minten.

**Writing – original draft:** Eline D'Haene, Senne Vandevelde, Bart Minten.

**Writing – review & editing:** Eline D'Haene, Senne Vandevelde, Bart Minten.

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
