## [Decision Letter · Decision Letter 0]

25 May 2021

PONE-D-21-11710

Fasting, Food and Farming: Value Chains and Food Taboos in Ethiopia

PLOS ONE

Dear Dr. D'Haene,

Thank you for submitting your manuscript to PLOS ONE. After careful consideration, we feel that it has merit but does not fully meet PLOS ONE’s publication criteria as it currently stands. Therefore, we invite you to submit a revised version of the manuscript that addresses the points raised during the review process.

Please address the minor revisions proposed by the independent reviewer.

We look forward to receiving your revised manuscript.

Kind regards,

Eric Fèvre

Academic Editor

PLOS ONE

Additional Editor Comments:

Please see the comments of the reviewer and address these.

Journal Requirements:

Reviewers' comments:

Reviewer's Responses to Questions

**Comments to the Author**

1. Is the manuscript technically sound, and do the data support the conclusions?

Reviewer #1: Yes

2. Has the statistical analysis been performed appropriately and rigorously? 

Reviewer #1: Yes

3. Have the authors made all data underlying the findings in their manuscript fully available?

Reviewer #1: Yes

4. Is the manuscript presented in an intelligible fashion and written in standard English?

Reviewer #1: Yes

5. Review Comments to the Author

Reviewer #1: Pge 3, Line 46

Needs to be qualified. Likely true in low income countries, but in upper middle income and high income countries with high meat consumption. “There is a clear link between high intake of red and processed meats and a higher risk for heart disease, cancer, diabetes and premature death” Hu, Harvard

Page 4 line 82 – describe the fasting, is it like Ramadan where you eat at night? What, how much and when are they allowed to eat during the fasting.

Page 5, line 105-107

Are we using ASF and dairy interchangeably. ASF can include poultry which is not dairy. Perhaps an explanation of what is ASF in Ethiopia. What is the normal diet – we have no idea so far

Page 7 line paragraph 1 What is the dietary pattern and calories consumed during fasting vs nonfasting for the adults and for the chidren

Page 9 line 222

Is there any IRB in Ethiopia? Perhaps at the university? Was any consent sought from any government entity, like the Ministry of Health or Agriculture?

Line 331-337: Could this be a cohort effect where the pattern is changing, and younger persons who are more exposed, are less adherent. This might also explain the difference in these estimates vs previous studies.

Line 483 – Still no overall depiction of calories and dietary patterns during fasting and non-fasting for adults and children, remote and non-remote

6. PLOS authors have the option to publish the peer review history of their article (what does this mean?). If published, this will include your full peer review and any attached files.

Reviewer #1: **Yes: **T. Alafia Samuels

---

## [Author Response · Author response to Decision Letter 0]

9 Jul 2021

Dear Reviewer, Dear Prof. Dr. Alafia Samuels

We would like to thank you for your valuable comments on our manuscript. The constructive feedback has been very helpful in guiding our revision. We think the modifications have improved the paper.

In this reply letter, we provide a detailed account of the changes we have made and we explain how we have addressed each comment. Next to this letter, we also provide you with the revised manuscript. We have indicated the changes using "Track Changes”. 

We hope you are satisfied with the way we have addressed these comments. We would be glad to answer any further queries, and adjust the paper if more questions or suggestions would arise.

Kind regards,

the authors

---

## [Editor Report · Decision Letter 1]

2 Nov 2021

Fasting, Food and Farming: Value Chains and Food Taboos in Ethiopia

PONE-D-21-11710R1

Dear Dr. D'Haene,

We’re pleased to inform you that your manuscript has been judged scientifically suitable for publication and will be formally accepted for publication once it meets all outstanding technical requirements.

Kind regards,

Eric Fèvre

Academic Editor

PLOS ONE

Additional Editor Comments (optional):

Many thanks for your responses to the reviewer, which have been shared and now now approved.
---

## [Editor Report · Acceptance letter]

15 Nov 2021

PONE-D-21-11710R1 

Fasting, Food and Farming: Value Chains and Food Taboos in Ethiopia 

Dear Dr. D'Haene:

I'm pleased to inform you that your manuscript has been deemed suitable for publication in PLOS ONE. Congratulations! Your manuscript is now with our production department. 

Kind regards, 

on behalf of

Prof. Eric Fèvre 

Academic Editor

PLOS ONE